# Pathophysiology of the Different Clinical Phenotypes of Chronic Inflammatory Demyelinating Polyradiculoneuropathy (CIDP)

**DOI:** 10.3390/ijms23010179

**Published:** 2021-12-24

**Authors:** Edyta Dziadkowiak, Marta Waliszewska-Prosół, Marta Nowakowska-Kotas, Sławomir Budrewicz, Zofia Koszewicz, Magdalena Koszewicz

**Affiliations:** 1Department of Neurology, Wroclaw Medical University, Borowska 213 Str., 50-556 Wroclaw, Poland; edyta.dziadkowiak@umw.edu.pl (E.D.); marta.nowakowska-kotas@umw.edu.pl (M.N.-K.); slawomir.budrewicz@umw.edu.pl (S.B.); magdalena.koszewicz@umw.edu.pl (M.K.); 2Faculty of Architecture, Wroclaw University of Science and Technology, 50-317 Wroclaw, Poland; zosia@koszewice.pl

**Keywords:** chronic inflammatory demyelinating polyneuropathy, CIDP, phenotypes, treatment

## Abstract

Chronic inflammatory demyelinating polyneuropathy (CIDP) is the most common form of autoimmune polyneuropathy. It is a chronic disease and may be monophasic, progressive or recurrent with exacerbations and incomplete remissions, causing accumulating disability. In recent years, there has been rapid progress in understanding the background of CIDP, which allowed us to distinguish specific phenotypes of this disease. This in turn allowed us to better understand the mechanism of response or non-response to various forms of therapy. On the basis of a review of the relevant literature, the authors present the current state of knowledge concerning the pathophysiology of the different clinical phenotypes of CIDP as well as ongoing research in this field, with reference to key points of immune-mediated processes involved in the background of CIDP.

## 1. Introduction

Chronic inflammatory demyelinating polyneuropathy (CIDP) is the most common form of autoimmune polyneuropathy. It is a chronic disease and may be monophasic, progressive or recurrent. Patients with CIDP were first described by Hermann Eichhorst, a German neurologist working in Switzerland, in 1890 [1]. In the 1950s, a clinical concept of inflammatory polyneuropathies responding to treatment with corticosteroids was developed. In the 1970s, chronic recurrent inflammatory polyneuropathy was described as a separate disease entity, the present name was given in the 1980s [2,3], and the English name of CIDP was introduced by Dyck et al. [4]. Dyck et al. also introduced the basic clinical features of the disease. Since then, many descriptions of varieties of this neuropathy have appeared, as well as numerous proposals of diagnostic criteria [5,6].

The prevalence of CIDP is estimated at 1.2–8.9 per 100,000 people. The disease can appear at any age, more often around the age of 50; 10% of CIDP cases are children (rarely under the age of 1) [7]. Pre-symptoms, such as immunization with a foreign protein (e.g., vaccination), infectious diseases, are much less common than in Guillain-Barré syndrome [8]. 

Clinically, CIDP presents as a progressive symmetrical limbs’ paresis that affects both the distal and proximal parts. The essence of CIDP is a selective involvement of peripheral nerves, plexuses or nerve roots. Limb weakness is accompanied by paraesthesia and a decreased sense of touch, pain and temperature. Impairment of deep sensation can lead to imbalance. Occasionally, cranial nerve involvement is observed. Tendon reflexes are weakened or suppressed in all limbs [6,7,8]. CIDP symptoms develop fully, usually within 2 months. In some patients, the course may be more rapid (acute—onset CIDP)—the full clinical picture may develop within a few weeks, which may resemble the course of the Guillain-Barre syndrome [6,9]. The disease may be monophasic, progress continuously or periodically regress and recur [10,11].

The current criteria for the diagnosis of CIDP have been developed by the European Federation of Neurological Societies (EFNS) in cooperation with the Peripheral Nerve Society (PNS) in 2021 [6]. According to these criteria, CIDP should be suspected in any case of progressive, symmetrical or asymmetric polyradiculoneuropathy, the symptoms of which recur or worsen within 2 months. Symptoms strongly suggesting CIDP include: paresis of the distal limbs, abolition of tendon reflexes, sensory disturbances, and decreased vibration sensation. In patients with such suspicion, it is necessary to perform a nerve conduction test. The examination should include at least four peripheral nerves [6,7,9]. According to the criteria mentioned above, stimulation of the median, ulnar, peroneal and sural nerves should be performed on the same side. If the criteria are not met, the same nerves should be examined on both sides [6].

The electrodiagnostic examinations may be supplemented by examination of the cerebrospinal fluid and Magnetic Resonance Imaging (MRI) of the roots and nerve plexuses. Peripheral nerve biopsy does not give a definite confirmation/exclusion of the CIDP diagnosis [6,12,13].

The typical form of CIDP is a sensorimotor form with a chronic onset, symmetrical symptoms distribution, usually more proximal than distal [14]. In approximately 16% of patients with typical CIDP, the onset is sudden (acute-onset CIDP), with the greatest intensity of symptoms in less than 2 months [12,14]. The atypical forms of CIDP or CIDP variants include: the form with predominant sensory symptoms (sensory), including the chronic immune sensory polyradiculopathy (CISP) form, the form with the predominance of motor symptoms (motor CIDP), including the axonal motor form (MAMA, multifocal acquired motor axonopathy), with predominance of symptoms in the distal segments (DADS, distal acquired demyelinating symmetric neuropathy), asymmetric form (multifocal acquired sensory-motor neuropathy [MADSAM, multifocal acquired demyelinating sensors and motor neuropathy]) and focal (with the involvement of one or more nerves in one limb, with symptoms of involvement of the brachial or lumbosacral plexuses) [6,7,13]. Currently, it is assumed that the phenotypes of CIDP may change, that is, asymmetric forms may transform into the more typical symmetrical form of CIDP (Table 1) [6].

## 2. Methods

### 2.1. Search Strategy

The authors conducted a literature search focused on the topic of immunological characteristic of various phenotypes of CIDP. The search engines PubMed via MEDLINE and Google Scholar were used from the beginning of 2010, until 31 October 2021. Reviews and research studies, classified according to their relevance, were initially included, with subsequent exclusion of conference abstracts and papers written in languages other than English. In addition, reference lists from the eligible publications were searched for their relevance to the topic. With the uses of keywords as follows: chronic inflammatory demyelinating polyneuropathy, CIDP, anatomy, physiology, phenotypes, immunology, inflammatory process, demyelination process, nodal and paranodal antibodies, and atypical form. In addition to using singular key words, to find the most relevant records, the authors used PubMed Advanced Search Builder as well. The advanced queries used were as follows: (((CIDP) AND (phenotypes)) AND (immunology))); ((chronic inflammatory demyelinating polyneuropathy) OR (CIDP)) AND (((inflammatory process) OR (immunology)) OR (atypical form))). To find the most relevant paper, two analysts (ED and MWP) were working separately by screening the search engines.

### 2.2. Data Extraction

As a result, 743 records were identified and screened separately by ED and MWP. Each of the analysts prepared their own list of records identified as relevant to the study. Then, these record lists were double read by both of them, and 110 abstracts were found to be relevant. Then, full text manuscripts were acquired. Then, all articles were read independently by both analysts. 

### 2.3. Qualitive Analysis and Synthesis

Each analyst worked independently and prepared their own list of relevant full-text manuscripts. Both lists were compered, and by discussion, 83 publications were found to be the most relevant to the study and included in this review. 

## 3. Anatomy

### 3.1. The Structure of Myelin Sheaths 

Two critical physical parameters influence the speed of conduction in the nervous system: the axon’s diameter and the axon membrane’s resistance to ion leakage out of the cell. The larger the axon diameter or the more leak-resistant the membrane, the faster an action potential will move. The myelin sheath, the greatly extended and modified plasma membrane, is wrapped around the nerves’ axons, isolating them. The ion channels are clustered in Ranvier nodes, holding the action potential in the nodes regions. It reduces the space and time requirements for the action potential propagation continuous to saltatory conduction. Structures located in nodes and paranodal region are important in numerous pathologies of the peripheral nervous system, which are associated with dysfunction of nodal gangliosides 170,171 [15,16], neurofascin family, Caspr1, and contactin1 [17]. 

The formation of Ranvier’s node is associated with the clustering of sodium channels near the edge of the forming myelin sheath. This process has been observed in the peripheral nervous system, studying Schwann cells [18] and in the central nervous system, where oligodendrocytes are involved in myelin formation [19]. Clustering of channels is mainly dependent on direct contact with glial cells, or probably with different factors secreted by various subpopulations of glial cells [20,21,22]. The structure of Ranvier’s node in the peripheral system may differentiate following the contact with Schwann cells via microvilli [23]. In the central nervous system, this role is more likely played by astrocytes and processes of glial progenitor cells [24,25]. 

Schwann cell microvilli surround an area that can be divided into the node of Ranvier (NOR), the paranodal junction (PNJ), and the juxtaparanodal region (JXP). All of them are covered by the basal lamina (BL) [26]. The integrity of these domains depends on specific cell adhesion molecules (CAMs), and the integrity of neighboring domains. 

In turn, structures relevant to the pathophysiology of CIDP can be distinguished in each of these regions (for NOR region and PNJ—Figure 1) [26]. 

### 3.2. The Structure of Ranvier’s Node and Paranodal Region

Ranvier’s node is an area enriched in sodium channels (NaCh) and diverse potassium channels (TRAAK, TREK, Kv7.2/Kv7.3 and Kv3.1b) [27,28,29]. These channels are associated with ankyrin G and ankyrin R proteins, which belong to a family of proteins that link membrane proteins to the spectrin-actin complex cytoskeleton [28,30]. In addition, through the ankyrin proteins, these channels are associated with neurofascin 186 (NF186) and with a protein belonging to the immunoglobulin superfamily—NrCAM [31]. The other neurofascin isoform (NF140) is expressed mainly during the formation of the node of Ranvier, NF186 is involved in sodium channel clustering. The expression of the NF140 is also found in demyelinated white matter of the brain in multiple sclerosis [32].

Several transmembrane proteins of Schwann cells are expressed in the node of Ranvier, such as NrCAM, gliomedin, dystroglycan (αDG and βDG), syndecan 3 and 4 (Syn3 and Syn4) and protein M6B [33,34,35,36]. 

In the PNJ in the central and peripheral nervous systems, the neurofascin protein (NF155) and myelin-associated glycoprotein (MAG) are staged on the glial cell side. In Schwann cells NF 155 binds, with the help of protein 4.1G, to the ankyrin B (AnkB) and further to the proteins of the cytoskeleton: spectrin β2 and α2 [37]. Axonal adhesion complex composed of contactin and Caspr1 (Contactin associated protein-1) participates in junction formation on the axon side. Through Caspr1, which binds to protein 4.1B, contact with spectrins is provided. 

The paranodal region full of axo-glial junctions play a role in stabilization of nodes probably by acting as diffusion barriers, segregating ion channels at the node from those in the juxtaparanodes [38] and limiting the lateral diffusion of the nodal complex [39,40].

The axonal membrane contains several isoforms of voltage-gated potassium channels (Kv1) at the JXP [41]. Potassium channels associate with Caspr2, and Caspr2/K+ channel complexes are formated. On both the axon and glial cell sides in juxtaparanodes the presence of TAG-1 (a cell adhesion molecule, also named Contactin2/TAG-1) was stated [42,43]. The localization of Kv1 channels at the JXP strongly depends on both Caspr2 and TAG-1 presence [44]. 

Protein 4.1 families also play a crucial role in the integrity of the JXP, e.g., 4.1G targets and stabilizes the glial cell adhesion molecules (Necl1, Necl2, Necl4) [45]. Transmembrane receptors ADAM22 and ADAM23 integrate with LGI family proteins and recruit PSD-93 (postsynaptic density protein 93) and PSD-95 (postsynaptic density protein 95) [46,47]. The last one is a member of the membrane-associated guanylate kinase (MAGUK) family and bind to Kv1 channels, without the Caspr2 [47].

### 3.3. Nodal and Paranodal Associated Antibodies 

Antibodies against nodal and paranodal proteins: NF155, NF140, NF186, CNTN1, and Caspr1 have been described in CIDP [48,49,50,51]. They belong to immunoglobulin G4 (IgG4), G3 (IgG3) or G1 (IgG1) classes [49,50,51]. They are thought to be responsible for differences in the clinical picture and response to treatment in CIDP patients. 

The standard treatment, considered an autoimmune origin of CIDP, is based on the administration of corticosteroids, immunoglobulins, plasma exchange and immunosuppressive therapy [6]. The subgroup of patients with disruption of axoglial junctions in the node/paranode region tends to show tremor, ataxia, cranial nerve involvement and poor response to intravenous immunoglobulin (IVIg) [19,51,52,53]. 

Patients without defined presence of antibodies (seronegative) CIDP respond to immunotherapy in 60–80% of cases, while as much as 80% of patients with nodal/paranodal antibodies have poor response to the therapy [50,52]. IVIg in seronegative CIDP patients might inhibit the complement pathway and modulate the Fc receptors on macrophages, promoting remyelination, although some data are contrary to this theory [54].

The exact mechanism of nodal and paranodal antibodies remains unclear, and probably there are some discrepancies among distinct classes [52]. The poor response to IVIg in some IgG4 positive patients (seropositive CIDP) may be due to the lack of complement-mediated inflammatory cascade and macrophage-mediated demyelination [55,56]. In this case, rituximab, a monoclonal antibody against CD20, important in the B cell depletion process, seems to be more effective [57]. The latest observations show that rituximab has good effects in seropositive CIDP patients, although effectiveness decreases with the disease duration and degree of axonal damage [58,59,60]. Randomized clinical trials to determine the effectiveness and safety of rituximab in CIDP patients, especially in the subgroup with nodal/paranodal antibodies are postulated. Patients with nodal and paranodal antibodies tend to present atypical forms of CIDP (Table 2). 

## 4. Pathogenesis 

### 4.1. Inflammatory Process

CIDP is a form of chronic neuropathy that is presumably caused by heterogeneous immune-mediated processes. The exact mechanisms of neuropathy in classical macrophage-induced demyelination remain unclear despite the long-standing recognition of this process in CIDP [1,2,3,4].

Classical macrophage-induced demyelination is basic in the pathogenesis of CIDP and is found in some patients in every major subtype, including typical CIDP, DADS, MADSAM and purely sensory subtypes. Early ultrastructural studies using biopsy specimens from patients with CIDP have demonstrated the stripping of morphologically normal myelin lamellae by cytoplasmic processes of macrophages [65,66]. Myelin damage is mainly provoked by complement—depended antibodies’ activation. The recent studies have shed light on antibody-dependent phagocytosis by macrophages without participation of complements. However, direct association between specific autoantibodies and macrophage-induced demyelination has not been reported. Electron microscopic examination of longitudinal sections of sural nerve biopsy specimens suggested that macrophages recognize specific sites of myelinated fibers as the initial target of demyelination. The site that macrophages select to initiate myelin breakdown is located around the nodal or internode regions. Hence, it seems that the components system distinguishing between the nodal and paranodal regions plays a pivotal role in the behavior of macrophages that initiate phagocytosis of myelin [67].

### 4.2. Demyelination Process

Recent studies have led to the concept of nodopathy or paranodopathy. The mechanisms initiated by autoantibodies against paranodal junction proteins (neurofascin 155 and contactin 1) have been demonstrated. Paranodal dissection resulting from the attachment of immunoglobulin G4 (IgG4) at paranodal junctions and the absence of macrophage-induced demyelination are characteristic pathologic features in patients with such antibodies [67]. Antibodies against proteins of the node of Ranvier and the paranodal regions have been demonstrated in approximately 10% of patients diagnosed with CIDP, showing atypical clinical phenotypes and inadequate response to standard CIDP treatment. Antibodies, are directed against various cellular adhesion proteins located in or adjacent to the node of Ranvier [13,68].

Differences in lesion distribution and repair processes by Schwann cells may determine differences between subtypes. It is suggested that in typical CIDP there is preferential involvement of proximal and distal nerve segments, whereas in MADSAM there is a marked involvement of central nerve segments. These findings suggest that humoral rather than cellular immunity predominates in the former, as the nerve roots and neuromuscular junctions are devoid of the blood-nerve barrier [1,13,59].

Studies have also compared the immunological profile of CIDP patients with IgG4 anti-neurofascin 155 (NF155) antibodies (NF155^+^ CIDP) with patients lacking anti-NF155 antibodies (NF155^-^ CIDP). A macrophage-induced demyelination was not observed in NF155^+^ CIDP [68]. In NF155^+^ CIDP, the downregulation of IL1β is consistent with the absence of macrophage-mediated demyelination. Lymphocytes Th2 and IL 13 cytokines downregulate IL1β. Thus, it is possible that overrepresented Th2 (type 2 helper T cell) cytokines play a critical role in inducing IgG4 autoantibodies via the effects on IL4/IL13/IL10 and spinal root inflammation or by downregulation of IL1β and macrophage functions [8,69]. A decrease in anti-inflammatory cytokine, IL-1 receptor antagonist (IL-1ra) was common in both NF155^+^ and NF155^−^ CIDP, but more marked in NF155^+^ CIDP. IL-1ra is a representative anti-inflammatory cytokine. Pronounced decrease in IL-1ra levels in NF155^+^ CIDP patients may also contribute to severe spinal root inflammation. Intrathecal upregulation Th2 cell cytokines is characteristic for IgG4 NF155^+^ CIDP, while type 1 helper T cell (Th1) cytokines are increased in CIDP regardless of the presence or absence of anti-NF155 antibodies. The findings suggest that overproduction of Th2 cell cytokines is unique to NF155^+^ CIDP [69].

Based on the recent studies, CIDP may be related to immunoglobulin G4 (IgG4) reactivity. Immunoglobulin IgG4 is not capable of activating the classical complement pathway or forming immune complexes. Subclasses lacking the ability to activate complement can interact with other immunoglobulin subclasses and activate complement via the lectin pathway in the mouse models. IgG4 can also act as a neutralizing or blocking antibody, protecting the organism from severe allergic reactions caused by food or environmental allergens. In the response to Th2-dependent cytokines such as interleukins (IL)-4, -5, -10 and -13 and transforming growth factor β (TGF-β) eosinophilia activation can occur, IgG4 and IgE levels increase and fibrosis progresses. These cytokines, especially IL-10, are responsible for allergic symptoms, eosinophilia, and increased IgE and IgG4 levels. It seems that the overexpression of Th2 lymphocytes and Treg lymphocytes and their dependent cytokines plays a major role in the pathogenesis of IgG4-mediated diseases. The role of innate immunity in the pathogenesis of IgG4-RD is also under consideration. Peripheral blood innate immune cells, such as plasmacytoid dendritic cells and monocytes isolated from patients with IgG4-RD, promote IgG4 production by B cells. Activation of the innate immune response by microbe- and/or damage-associated molecular patterns stimulates production of type I interferon and B cell-activating factor by innate immune cells and results in IgG4 production by B cells. Macrophage and basophil cell dysfunction and overexpression of various cytokines including BAFF (TNF superfamily member) and APRIL (a proliferation-inducing ligand) have been demonstrated [70,71,72].

#### 4.2.1. The Role of Autoantibodies against Nodal or Paranodal Proteins

Neurofascin is crucial in constructing and maintaining the nodes of Ranvier. The mature nervous system predominantly expresses a neuronal isoform, NF186, and a glial isoform, NF155, whereas immature neurons express NF180 and NF166. Glial NF155 is expressed at paranodal loops of Schwann cells in the PNS and in oligodendrocytes in the CNS. Glial NF155 acts as a cell adhesion molecule, interacting with axonal CNTN1 and CASPR1 [61,73,74,75,76,77]. Individual autoantibodies are implicated in unique features; consequently, CIDP associated with these nodal/paranodal autoantibodies is now recognized as an autoimmune nodopathy or paranodopathy.

The CNTN1-Caspr1-NF155 complex is required to maintain paranodal architecture and to maintain myelin insulation of the axon for nerve impulses propagation along myelinated axons. In patients with CIDP IgG1 and IgG4 subclasses autoantibodies are directed against CNTN1, and they can alter the paranodal architecture. IgG4 antibodies have a potential to block the NF155-CNTN1 interaction. Autoantibodies against NF155 (as well as against CNTN1 and possibly Caspr1) cause demyelinating lesions in both the CNS and PNS, termed combined central and peripheral demyelination (CCPD). However, there are some differences between individual patients. Patients with NF155 antibody-positive CIDP have a distinct clinical phenotype, which is defined by low-frequency, high-amplitude tremor, sensory ataxia and poor response to IVIG. Additionally, the anti-NF155 antibody-positive form of CIDP is exceptional in that it is associated with a high incidence of subclinical demyelinating lesions in the central nervous system [75,78,79]. At least two pathogenetic mechanisms have been proposed for NF155-IgG4 CIDP: a blocking mechanism that prevents the interaction between CNTN1-/Caspr1 and NF155 and disrupts the paranodal structure, and a second induction of NF155 clustering and removal from the cell surface by an unidentified mechanisms, in the apparent absence of a blocking function [62,63,78,79].

Neurofascins -140 and -186 (NF140, NF186), also expressed on motoneuron paranodes, seem to be the main target of autoantibodies in some CIDP patients [73,74,75,76,77]. Kira et al. presented a hypothetical model of mechanism of NF155+ CIDP and combined central and peripheral demyelination (CCPD). NF155 peptides are presented by the DRB1*15:01/DRB1*15:02 and/or DQA1*01:02-DQB1*06:02/DQA1*01:03-DQB1*06:01 complex to naive T cells, initiating Tfh2/Th1 cell differentiation. Tfh2 (follicular helper T type 2) cells produce interleukins IL4/IL13/IL10, which induce IgG4 class switching. IgG4 anti-NF155 antibodies penetrate the nerve terminal and nerve roots where the blood-nerve barrier is absent or leaky. Anti-NF155 antibodies disrupt the interaction between NF155 and the CNTN-1/Caspr1 complex in the paraganglia, leading to the detachment of Schwann cell terminal loop from axons. Activated Th2 and Th1 cells induce inflammation in spinal roots, leading to nerve root hypertrophy and sometimes to oval periventricular lesions in the central nervous system. Overproduction of IL13 decreases IL1β production, which inhibits macrophage activation and recruitment. Cranial nerves such as the optic nerve, oculomotor nerve, trigeminal nerve and facial nerve are also damaged by anti-NF155 antibodies and probably by activated Th2/Th1 cells (Figure 2) [69,74].

#### 4.2.2. The Role of Antibodies against the Hemi-Node-Type Region

Motor neurons have heterogeneous axon initial segments (AISs), which underlie different spiking properties. Duflocq et al. identified a hemi-node-type organization in all α-motor neurons, with a contact-related protein (Caspr)+ paranode and a Caspr2+ and Kv1+ paranode compartment, identified as para-AIS and juxtapara (JXP)-AIS, adjacent to the AIS where the myelin sheath begins, which might limit some AIS plasticity [64]. Protein 4.1B plays a key role in ensuring the proper molecular compartmentalization of this hemi-node-type region [64,80,81].

There are only a few publications on CIDP patients with IgG4 subclass antibodies against Caspr1, therefore antibodies against other protein located on motoneuron paranodes or complexed Caspr1/CNTN1 should be considered [62,63,79].

## 5. Conclusions

CIDP is an acquired autoimmune neuropathy the pathogenesis that is not yet well understood. The involvement of autoreactive T lymphocytes, B lymphocytes, complement components, inflammatory chemokines and cytokines, antibodies to various glycoprotein and glycolipid nerve structures has been confirmed in the development of CIDP. The discovery of autoantibodies against the proteins of the node of Ranvier and the paranodal region in patients with atypical CIDP confirms the pathogenetic variety. The patients with these antibodies are mostly young, and the course of the disease is sudden, with the development of significant disability and a poor response to immunoglobulin treatment, but a potentially positive response to rituximab.

Advances in research into the immunopathogenesis of CIDP will contribute to the correct diagnosis of this neuropathy and the application of the effective treatment.

## Figures and Tables

**Figure 1 ijms-23-00179-f001:**
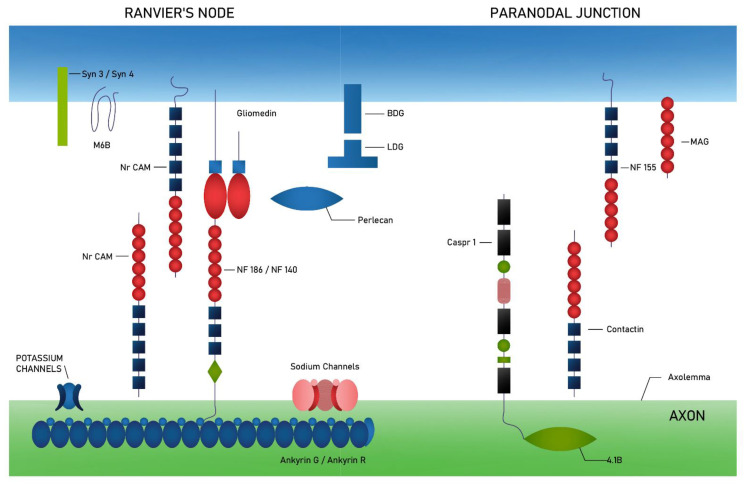
Schematic representation of the proteins that make up the Ranvier’s node and paranodal junction. On the axonal side, the Ranvier’s node consist of sodium channels (NaCh) and diverse (mechanically and voltage-gated) potassium channels (K channels), which are associated with ankyrin G and ankyrin R proteins—linking membrane proteins to the spectrin-actin complex cytoskeleton—and at the same time with neurofascin 186 (NF186) or neurofascin 140 (NF 140). Adhesion complex in paranodal region composes of contactin and contactin associated protein-1 (Caspr 1), which bind to protein 4.1B. Schwann cells express in Ranvier’s node several transmembrane proteins such as protein belonging to the immunoglobulin superfamily (NrCAM), dystroglycan (αDG and βDG), syndecan 3 and 4 (Syn3 and Syn4) and protein M6B. In paranodal junction, myelin-associated glycoprotein (MAG) and neurofascin 155 (NF 155) are expressed [26].

**Figure 2 ijms-23-00179-f002:**
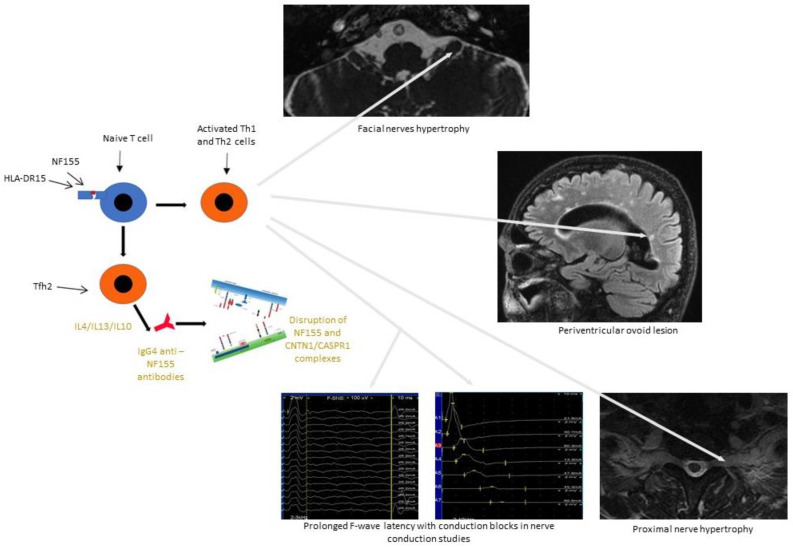
Chronic inflammatory demyelinating polyneuropathy (CIDP) with anti-NF155 antibodies—the possible mechanism of peripheral and central lesion. After a presentation of NF155 peptides by HLA-DR15 or HLA-DQA1 complex to naive T cells, the differentiation of Tfh2/Th1 cells starts. Tfh2 cells produce interleukins such as IL4/IL10/IL13, which induce IgG4 class switching. IgG4 anti-NF155 antibodies move into places where the blood–nerve barrier is absent or leaky, such a nerve terminal and nerve roots. They disrupt the interaction between NF155 and the CNTN-1/Caspr1 complex at the paranode. Activated Th1 and Th2 cells induce inflammation at the spinal roots, causing nerve roots hypertrophy, affection of the cranial nerves and/or periventricular ovoid lesions in the central nervous system.Scale bar of the MRI images is 1:6.

**Table 1 ijms-23-00179-t001:** Phenotypes of CIDP [6].

Typical CIDP	CIDP Variants
All the following: Progressive or relapsing, symmetric, proximal and distal muscle weakness of upper and lower limbs, and sensory involvement of at least two limbs Developing over at least 8 weeks Absent or reduced tendon reflexes in all limbs	One of the following, but otherwise as in typical CIDP (tendon reflexes may be normal in unaffected limbs): Distal CIDP: distal sensory loss and muscle weakness predominantly in lower limbs Multifocal CIDP: sensory loss and muscle weakness in a multifocal pattern, usually asymmetric, upper limb predominant, in more than one limb Focal CIDP: sensory loss and muscle weakness in only one limb Motor CIDP: motor symptoms and signs without sensory involvement Sensory CIDP: sensory symptoms and signs without motor involvement

Abbreviation: CIDP, chronic inflammatory demyelinating polyradiculoneuropathy.

**Table 2 ijms-23-00179-t002:** IgG4 antibodies against nodal/paranodal proteins and reaction to treatment [13,61].

Antibodies	Frequency in CIDP Patients	Localisation	IvIg	Corticosteroids	Plasma Exchange	Rituximab
NF155	1–21% [19,62,63]	Paranodal	Poor response	Partial response	Potentially good response	Potentially good response
CNTN1	0.7–8% [19,52,62]	Paranodal	Poor response	Partial response	Partial response	Potentially good response
NF140/NF186	2–5% [48,49]	Nodal	Partial response	Partial response	Potentially good response	Potentially good response
Caspr1	0.2–3% [19,64]	Paranodal	Poor response	Partial response	Partial response	Potentially good response

## Data Availability

The data presented in this study are available upon request from The corresponding author. The data are not publicly available.

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
