# Peer review of "Pathophysiology of the Different Clinical Phenotypes of Chronic Inflammatory Demyelinating Polyradiculoneuropathy (CIDP)"

_ijms, 2021, doi:10.3390/ijms23010179_

Round 1

Reviewer 1 Report

In the paper, the authors review the current state of knowledge concerning the pathophysiology of the different clinical phenotypes of CIDP. The review is generally well-organized and focused, however there are some issues that should be addressed to improve the manuscript.

-The authors indicate, both in the title and in the abstract, that “therapeutic implications” will be discussed. Almost no discussion is done on this point. The authors should modify the abstract and title (as it is misleading) or, should significantly expand the review to include current and novel therapeutic avenues aimed at treating the different forms of CIDP.

-The “Methods” should be accompanied by a graphical representation of the workflow, indicating the number of papers screened, the number of papers included  in the review and the number (and reasons) for exclusion of papers (if any).

-Table 2. It is not clear how the indicated frequencies are generated. Have the authors performed a meta-analysis of the results presented in the references? This is an important point. The methods for the meta-analysis and statistical significance should be clearly stated.

Author Response

Authors' response to the Reviewer comments

Dear Reviewer,

            Thank you very much for your review. All comments and suggestions from the reviewers have been included in the manuscript. English language and style was check by native speaker. Point-by-point responses:

  1. The authors indicate, both in the title and in the abstract, that “therapeutic implications” will be discussed. Almost no discussion is done on this point. The authors should modify the abstract and title (as it is misleading) or, should significantly expand the review to include current and novel therapeutic avenues aimed at treating the different forms of CIDP.

We have modified the title and the abstract. We removed the treatment section that did not actually cover the entire paper.

  1. The “Methods” should be accompanied by a graphical representation of the workflow, indicating the number of papers screened, the number of papers included in the review and the number (and reasons) for exclusion of papers (if any).

We supplemented the method section with search strategy, data extraction and qualitive analysis and synthesis. It seems that the description is so short and legible that there is no need to present it graphically.

  1. Table 2. It is not clear how the indicated frequencies are generated. Have the authors performed a meta-analysis of the results presented in the references? This is an important point. The methods for the meta-analysis and statistical significance should be clearly stated.

Only the few studies in which this frequency was assessed are selected in Table 2. Frequencies are listed from the lowest to the highest available in these works. No meta-analysis of the results and statistical analysis was performed due to the small sample of the observed results and the heterogeneous methodology of the observations.

Reviewer 2 Report

Dziadkowiak et al. present an intersting review about the state of knowledge of CIDP. The paper is well written and cover all the aspects of this complex pathology.

Point 1:

Fig 1 the schematic representation seem a drawing. I suggest the authors to be more "scientists" and not "painters"

Please rewrite line 93-94. The meaning is not very clear .

Author Response

Authors' response to the Reviewer comments

Dear Reviewer,

            Thank you very much for your review. All comments and suggestions from the reviewers have been included in the manuscript. English language and style was check by native speaker. Point-by-point responses:

1. Figure 1 the schematic representation seem a drawing. I suggest the authors to be more "scientists" and not "painters"

We corrected this figure.

2. Please rewrite line 93-94. The meaning is not very clear.

We rewrite this lines.

Hopefully, the revised version of the manuscript, considering the above issues, would be found suitable for publication.

Authors

Round 2

Reviewer 1 Report

Authors have addressed the points raised in the first round of revision